# β-HPV 8E6 Attenuates ATM and ATR Signaling in Response to UV Damage

**DOI:** 10.3390/pathogens8040267

**Published:** 2019-11-26

**Authors:** Jazmine A. Snow, Vaibhav Murthy, Dalton Dacus, Changkun Hu, Nicholas A. Wallace

**Affiliations:** Division of Biology, Kansas State University, Manhattan, KS 66502, USA; jazasnow@ksu.edu (J.A.S.); murthy.39@buckeyemail.osu.edu (V.M.); dacus@ksu.edu (D.D.); Chu1@ksu.edu (C.H.)

**Keywords:** genus beta human papillomavirus, ATM, ATR, nucleotide excision repair, translesion synthesis, cell cycle, UV

## Abstract

Given the high prevalence of cutaneous genus beta human papillomavirus (β-HPV) infections, it is important to understand how they manipulate their host cells. This is particularly true for cellular responses to UV damage, since our skin is continually exposed to UV. The E6 protein from β-genus HPV (β-HPV E6) decreases the abundance of two essential UV-repair kinases (ATM and ATR). Although β-HPV E6 reduces their availability, the impact on downstream signaling events is unclear. We demonstrate that β-HPV E6 decreases ATM and ATR activation. This inhibition extended to XPA, an ATR target necessary for UV repair, lowering both its phosphorylation and accumulation. β-HPV E6 also hindered POLη accumulation and foci formation, critical steps in translesion synthesis. ATM’s phosphorylation of BRCA1 is also attenuated by β-HPV E6. While there was a striking decrease in phosphorylation of direct ATM/ATR targets, events further down the cascade were not reduced. In summary, despite being incomplete, β-HPV 8E6’s hindrance of ATM/ATR has functional consequences.

## 1. Introduction

The human papillomavirus (HPV) family is made up of five genera (alpha, beta, gamma, mu and nu papillomaviruses), each containing a large number of individual HPV types [1,2]. The division into these groups is based on differences in the major capsid gene’s sequence [3,4]. Although all these genera contain members capable of causing disease, the alpha (α-HPV) and beta (β-HPV) genera have received the most research attention because of their connection and potential connection to cancer, respectively [5,6,7,8]. Certain members of the alpha papillomavirus genus are known to cause tumors in the anogenital tract and in the oropharynx [9]. These so-called high risk, or HR α-HPVs, cause tumors that are dependent on continued viral oncogene (HR α-HPV E6 and E7) expression, making it somewhat straightforward to connect their infections with tumorigenesis [10,11].

β-HPVs are far more difficult to definitively tie to malignancies but may contribute to non-melanoma skin cancer (NMSC) development in certain populations and potentially more broadly [12]. The difficulty in linking β-HPVs to NMSCs is that, unlike HR-α HPVs, they do not cause an infection that lingers in the tumor [13,14]. Their infections are more transient, lasting for months rather than decades like HR-α HPVs [15]. As a result, β-HPV infections are thought to act through a “hit and run” mechanism of oncogenesis [16,17]. This hypothesis holds that β-HPV infections act synergistically along with UV radiation to promote tumorigenic mutations that cause lasting changes to the cellular environment without being dependent on continued expression of β-HPV’s putative oncogenes (β-HPV E6 and E7) [18].

The “hit and run” hypothesis presents a challenge for epidemiologists that is further compounded by the fact that neither β-HPV infections nor NMSCs are rare. In fact, most people are sero-positive for at least one β-HPV and there are millions of NMSCs diagnosed each year [19,20,21]. The purposed link between β-HPV and NMSCs is best characterized in individuals with *Epidermodysplasia verruciformis* (EV), a genetic disease that is associated with an increased susceptibility to HPV infections, and in solid organ transplant recipients [22,23,24]. While a potential role in cancer warrants further investigation, the ubiquitous presence of β-HPV in our skin alone makes it important to further understand β-HPV biology.

Of β-HPV’s genes, β-HPV E6 is the most well characterized [25]. It alters multiple cell signaling pathways including MAML1, TGFβ, NOTCH and EGFR signaling [26,27,28]. It also binds and destabilizes the cellular histone acetyltransferase, p300 [29]. We have previously shown p300’s role as a transcription factor is required for robust expression of at least four essential DNA repair genes, including two essential repair kinases (ATM and ATR) [30,31,32]. Because of their position atop multiple repair pathways, we hypothesize that diminished ATM and ATR availability has a far-reaching impact on the ability of cells to protect themselves from UV radiation [33,34,35,36]. We test this hypothesis with a combination of in silico and in vitro analyses, specifically focusing on phosphorylation events that facilitate cell cycle regulation, nucleotide excision repair (NER), and translesion synthesis (TLS). NER is responsible for physically removing UV-induced DNA lesions and it has been shown that an essential protein, XPA, is stabilized by ATR phosphorylation [37,38]. The TLS pathway helps bypass UV lesions primarily through the TLS polymerase, POLη, which is regulated by ATR and p53 [39,40]. Finally, ATM and ATR control cell cycle progression via phosphorylation of CHK1 and CHK2 [41,42,43].

## 2. Results

### 2.1. ATR, ATM and p53 Have Distinct Transcription Effector Profiles

We have previously reported that β-HPV 8E6 decreases ATM and ATR abundance [30,31]. However, the extent that β-HPV 8E6 disrupts ATM and ATR signaling remains poorly defined. This motivated us to characterize the extent that β-HPV 8E6 alters ATM and ATR signaling pathways. As a first step, we performed an in silico screen of previously collected transcriptomic data featuring 877 different cell lines [44,45,46]. Cell lines with ATM/ATR expression with z-scores below −2 were considered to have low expression (28 and 22 cell lines respectively) and compared to the remaining cell lines. We focused our analysis on genes that belonged to two pathways involved in UV repair responses, namely nucleotide excision repair (NER) and translesion synthesis (TLS) as well as a few canonical ATR/ATM targets (BRCA1, CHEK1, CDC25A, and TP53) [47,48,49,50,51]. We were unable to perform this analysis for CHEK2, one of the most characterized ATM targets, as there was no data available in the transcriptomic data. Gene expression was plotted against statistical significance in volcano plots to highlight significant robust correlations (Figure 1).

As expected, ATM and ATR expression positively correlated with UV responsive and canonical target gene expressions. We complimented this approach by performing the reciprocal analysis of cells that had high expression of these kinases. Despite comparing different subsets of cell lines, we found similar correlations among ATM/ATR and UV-responsive gene expression (28 and 45 cell lines were observed to have excess ATM/ATR, respectively) (Appendix A). As a final computational effort, we compared expression profiles in cell lines segregated by the presence or absence of ATM/ATR truncating mutations (Appendix A). However, there was only a small number of cell lines with ATM/ATR truncating mutations available for analysis, therefore the significant gene expression changes are not as robust. This also supported the role of these kinases as broad regulators of UV responsive gene expression. For both low and high expression of ATM/ATR, the changes in all three gene expression groups were more strongly correlated with ATM expression than ATR expression (Figure 1, Appendix A).

Both capstone DNA repair kinases have multiple targets that they regulate primarily via phosphorylation [34]. p53 is preeminent among those targets for both ATM and ATR [52]. We have also previously shown that β-HPV 8E6 delays p53 stabilization in response to UV damage [53]. p53 is also a well-recognized transcription factor [54]. To gain some insight into how much of the expression profiles was the result of ATM/ATR signaling through p53, we characterized expression of ATM/ATR responsive genes in transcriptomic data from cell lines segregated by p53 expression (Table 1). This was distinct from both ATM and ATR expression profiles but shared some notable overlap. POLK had a positive correlation in all three settings suggesting that POLK may be regulated by ATM, ATR and p53. This includes the possibility that each regulates POLK expression independently as well as the possibility that ATM and ATR regulate POLK by stabilizing p53. We interpret these data as consistent with other reports describing distinct but overlapping roles for ATM, ATR and p53 in response to UV. However, not all of these changes remained significant when accounting for the false discovery rate associated with multiple comparisons (Appendix A). This prompted us to test our hypothesis using in vitro approaches.

### 2.2. β-HPV E6 Decreases ATM and ATR Activation

The correlations demonstrated by these data motivated us to interrogate β-HPV E6’s ability to decrease ATR/ATM signaling with in vitro systems, beginning with the ATM activation that occurs via autophosphorylation at Ser1981 (pATM). β-HPV infection occurs in keratinocytes, making them the preferred cell culture model. We used p300 abundance as a surrogate marker for β-HPV E6 expression to confirm expression of β-HPV E6 in primary keratinocytes (LXSN and β-HPV 8E6 HFKs). Since these cells are derived from patients, it was important to control for donor variability using lines derived from separate sources. To this end, we also tested our hypothesis in keratinocytes derived from a different donor and immortalized by exogenous hTERT expression (hTERT HFKs). Probing for the HA-tag on the β-HPV 8E6 expressed in hTERT HFKs provided proof of expression (Appendix A). Finally, hTERT HFKs mimic the telomerase activation that is a common in NMSCs providing insight into β-HPV E6 phenotypes in a relevant cellular environment [55].

β-HPV 8E6 decreased total and activated ATM in each of these cell lines (Figure 2A). This loss is seen more clearly in primary HFKs. The difference in activated ATM remained over 8 h after UV-induced ATM activation (Figure 2B,C, Appendix A). To determine if β-HPV 8E6 prevents ATM from phosphorylating its downstream targets, we probed for two canonical ATM targets associated with the DNA damage response, Ser1423 of BRCA1 (pBRCA1) and Thr68 of CHK2 (pCHK2) [43,56]. β-HPV 8E6 caused aberrations in both proteins’ reaction to UV (Figure 2B,C). pCHK2 accumulation and total CHK2 abundance were both decreased by β-HPV 8E6. pBRCA1 levels peaked higher, but this buildup was delayed, occurring several hours after they reach their maxima in vector control hTERT HFKs, demonstrating a delayed response similar to what we have reported for p53 for hTERT HFKs [30] (Figure 2B, Appendix A). We did not see this delayed peak in pBRCA1 occur in the primary HFKs (Figure 2C). β-HPV 8E6 also decreased p53 stabilization, but we cannot distinguish whether this is an ATM or ATR effect as both kinases stabilize p53 (Appendix A). To determine if β-HPV 8E6 changed ATM’s cellular position, we performed subcellular fractionation on cells before and after UV treatment. There were no robust differences in nuclear localization, suggesting that β-HPV 8E6 primarily impairs ATM activation via decreased expression and autophosphorylation. Interestingly, we did observe some changes in cytoplasmic localization between LXSN, vector control, and β-HPV 8E6 in untreated cells (Figure 2D). We then moved to ATR’s activation by autophosphorylation at Thr1989 (pATR). This typically occurs in response to single stranded DNA associated with replication stress [57]. β-HPV E6 decreased pATR in hTERT HFKs (Figure 3A). pATR levels increased in vector control HFKs and in β-HPV E6 over 6/8-h post UV. However, β-HPV E6 diminished pATR induction. This resulted in notably less pATR in cells with β-HPV 8E6 after UV (Figure 3B,C, Appendix A). Subcellular fractionation did not provide evidence that β-HPV E6 attenuated localization of pATR to the nucleus (Figure 3D).

### 2.3. β-HPV 8E6 Decreases Phosphorylation of ATR Target Proteins

We continued our characterization of β-HPV 8E6’s impact on UV signaling by examining ATR’s most established target, CHK1 [57,58,59]. ATR phosphorylates CHK1 at Ser345 (pCHK1) in response to replication stress and UV [59]. We saw a mild increase in a replication stress marker (RPA32 at Ser8 (pRPA32)) accompanying β-HPV 8E6 expression (Appendix A). In contrast, pCHK1 was decreased by β-HPV 8E6 (Figure 4A). To determine if CHEK1 transcription changed, RT-PCR was performed. β-HPV 8E6 caused a modest but non-significant decrease in CHEK1 mRNA consistent with our in silico data (Figure 1 and Figure 4B). Next, we probed pCHK1 and total CHK1 by immunoblot over a 6/8-h time course after UV. While UV elicited a sizable increase in pCHK1 within an hour of exposure in vector control cells, β-HPV 8E6 prevent all but a mild induction of pCHK1 (Figure 4C,D, Appendix A). These changes were independent of foreskin donor or hTERT activation.

CHK1 coordinates cell cycle progression at the G1-S boundary [60]. To determine if β-HPV 8E6 diminished phosphorylation downstream of CHK1 activation, we defined the phosphorylation status of CHK1 targets, beginning with CDC25A [42]. This dual–specificity protein phosphatase removes inhibitory phosphorylates from cyclin-dependent kinases, like CDK2, and other regulatory factors, like CDC2, allowing them to promote cell cycle progression. Highlighting the key role of CDC25A in tumorigenesis, it is frequently overexpressed in cancer cells and associated with poor cancer patient outcomes [61]. In response to UV, pCHK1 phosphorylates CDC25A increasing its proteasome-mediated turnover [42]. To our surprise, β-HPV 8E6 did not reliably change total CDC25A abundance or the protein’s phosphorylation at Thr507 (pCDC25A) (Figure 4E and Appendix A). β-HPV 8E6 caused inconsistent changes to pCDC2 (at Thr14) (Figure 4E and Appendix A). This could be explained either by β-HPV 8E6 not completely inhibiting ATR signaling or by redundant kinase activity. Despite not completely blocking ATR signaling, β-HPV 8E6 subtly changed cell cycle distribution, mildly increasing the proportion of cells in G1 (Figure 4F).

ATR promotes NER by phosphorylating and stabilizing XPA in response to UV [62,63]. We measured total XPA and XPA phosphorylation at Ser196 (pXPA) by immunoblot (Figure 5A). A small but insignificant decrease in XPA mRNA accompanied β-HPV 8E6 expression indicating that this decrease in abundance is not likely due to reduced transcription (Figure 5B). Next, we looked at pXPA and XPA by immunoblot over a 6/8-h time course after UV. We observed that pXPA was increased in LXSN, vector control, after cells were exposed to UV. However, in the presence of β-HPV 8E6, pXPA protein abundance was decreased even after exposure to UV (Figure 5C,D, Appendix A). While there was only a subtle decrease in pXPA in hTERT HFKs following UV, this decline is more visible in primary HFKs. For further validation, we observed XPA phosphorylation in a previously described osteosarcoma cell line expressing β-HPV 8E6. β-HPV 8E6 attenuated XPA phosphorylation in these cells (Appendix A). Consistent with our previous experiments, we found that β-HPV 8E6 did not affect XPA’s distribution in subcellular fraction experiments. (Figure 5E). However, immunofluorescence microscopy showed mild differences in XPA localization associated with β-HPV 8E6. Specifically, XPA remained dispersed throughout the cell. This contrasts with XPA’s localization in control cells where it is nearly exclusively nuclear. This data suggests that β-HPV 8E6 may have some ability to decrease nuclear localization of XPA after UV exposure. (Appendix A).

Immunoblot analysis also shows β-HPV 8E6 causes a decrease in POLη (Figure 6A). This reduction of POLη is more consistent in primary HFKs throughout the figure. In contrast, we did not find significant differences in the abundance of another TLS polymerase, POLκ (Figure 6A) [64]. β-HPV 8E6 marginally decreased POLH (gene for POLη) expression consistent with our in silico data, but this modest difference failed to reach statistical significance (Figure 1 and Figure 6B). Previous reports have shown that POLη stability is dependent on ATR phosphorylation during UV damage [39], leading us to speculate that β-HPV 8E6 altered POLη stability. β-HPV 8E6 does not change the abundance of other TLS proteins, such as RAD18 and ubiquitinated PCNA (Appendix A). Exposure to UV increased the abundance of POLη and POLκ in control cells (Figure 6C,D, Appendix A). While β-HPV 8E6 prevented POLη induction, POLκ rose more sharply after UV (Figure 6C,D, Appendix A). This may represent a compensatory response. Neither of these phenotypes were altered by hTERT activation and both were consistent among cells derived from different donors (Figure 6C,D, Appendix A). There are likely functional ramifications of the reduced POLη abundance as immunofluorescence microscopy demonstrated that β-HPV 8E6 reduced UV-induced POLη nuclear foci (Figure 6E, Appendix A).

## 3. Discussion

Pre-clinical studies and observations in immunocompromised people with NMSC support the role of β-HPV in NMSC development [65,66]. Yet, gaps in the molecular details of how β-HPV E6 changes the cellular environment remain. To address this challenge, we defined how β-HPV 8E6’s reduction of ATR and ATM impacted cell signaling in response to UV. This work expands the breadth of known UV-responsive pathways impaired by β-HPV E6 to include nucleotide excision repair and translesion synthesis (Figure 5 and Figure 6, respectively). Figure 7 details the β-HPV E6 induced changes to DNA repair and cell cycle regulation described throughout this paper.

First, we performed an in silico screen to find candidate genes likely to be regulated by ATM/ATR. We saw a positive correlation between ATM/ATR expression and the expression of UV responsive and canonical target gene expression (Figure 1). Moving from in silico analysis to in vitro, we show that β-HPV E6 decreases the autophosphorylation of ATM and ATR in primary and hTERT HFKs (Figure 2 and Figure 3, respectively). This led us to study proteins that are dependent on ATM- and/or ATR- phosphorylation. We saw that phosphorylation of two key ATM targets, BRCA1 and CHK2, was lessened by β-HPV E6 with and without UV exposure (Figure 2). Further studies will need to be done to determine the extent of which decreased phosphorylation of BRCA1 and CHK2 impacts their downstream signaling pathways.

Since ATM is mainly involved in double strand break repair rather than UV repair, we moved on to ATR and its downstream targets. Beginning with phosphorylation of one of the most characterized ATR targets, CHK1. We found that pCHK1 was diminished by β-HPV E6. Since CHK1 phosphorylation halts the cell cycle, we hypothesized that β-HPV 8E6 reduced cell cycle arrest after UV. To test this, we examined the phosphorylation of CHK1 targets. Surprisingly, there were no appreciable changes to the downstream proteins, CDC25A and CDC2, and only modest changes in the cell cycle profile in cells with β-HPV E6 (Figure 4). This partial inhibition may be attributed to β-HPV 8E6’s inability to completely eliminate p300, ATM or ATR. Alternatively, the phenotypes could be explained by the presence of secondary kinases capable of filling in for ATR. In either case, we suspect that there will be other examples where β-HPV 8E6’s inhibition of signaling pathways is significant but limited. As a result, the continued interrogation of abrogated signaling is both warranted and necessary.

β-HPV E6 was able to attenuate phosphorylation of XPA, a rate-limiting protein for NER. This may also result in altered subcellular localization of XPA, but our data do not support a strong conclusion in this regard (Figure 5 and Appendix A). With less pXPA protein present, we hypothesize that NER function will be attenuated in cells expressing β-HPV E6. This could lead to genomic instability due to the persistence of UV lesions that would typically be resolved by the NER pathway. It would also be advantageous for β-HPV. The virus is dependent on cellular replication but infects an anatomical site that is frequently exposed to UV. Failure to initiate NER could increase the likelihood that β-HPV infected cells continue to proliferate after UV damage, offering a more conducive environment for β-HPV replication. Clearly, future studies on β-HPV E6’s impact on NER are needed to better clarify the functional consequences of reduced XPA phosphorylation.

Lastly, we looked at POLη, the TLS polymerase most relevant for bypassing UV lesions. β-HPV E6 decreased POLη abundance with and without UV. POLη foci formation and localization were reduced with β-HPV E6 (Figure 6). The levels of another TLS polymerase, POLκ, were not by β-HPV E6. Thus, β-HPV E6 is not universally reducing the availability of TLS polymerases. Decreased POLη is expected to promote genomic instability by forcing TLS to rely on TLS polymerases less suited to bypass UV lesions. The experiments described here have a limited ability to test these ideas, but they justify further investigation.

Together these data better elucidate β-HPV E6’s manipulation of UV damage repair. While there were inconsistencies between primary HFKs and hTERT HFKs in our immunoblots, we put more emphasis on the phenotypes seen in the primary HFKs. Primary HFKs only grow for a limited time in culture and thus more closely mirror the typically transient β-HPV infection. Further, it would not be surprising if the differences were attributable to the known interactions between telomerase and DNA repair machinery [67]. However, lack of functional analysis limits the breadth of our conclusions. This will require a more detailed interrogation of cell cycle, NER, and TLS in the presence of β-HPV E6. Organotypic raft cultures and animal models could also provide biologically relevant insight in the monoculture experiments described here. Further, it would be beneficial to repeat these experiments in the presence of other β-HPV proteins (particularly β-HPV 8E7) and genes from other disease associated β-HPVs (e.g., HPV 38 and HPV 49).

## 4. Materials and Methods

### 4.1. Cell Culture

Primary human foreskin keratinocytes (HFKs) were isolated from neonatal human foreskins. HFKs were grown in EpiLife medium (Gibco, Billings, MT, USA) supplemented with calcium chloride (Gibco), human keratinocyte growth supplement (Gibco), and penicillin-streptomycin (Caisson, North Logan, UT, USA) or Keratinocyte Growth Medium 2 (Promocell, Heidelberg, Germany), Supplement Mix (Promocell), and penicillin-streptomycin (Caisson). hTERT human foreskin keratinocytes (hTERT HFKs), provided by Michael Underbrink (University of Texas Medical Branch, Galveston, TX, USA), are immortalized keratinocytes that constituently express telomerase (hTERT). hTERT HFKs were grown in EpiLife medium (Gibco) supplemented with calcium chloride (Gibco), human keratinocyte growth supplement (Gibco), and penicillin-streptomycin (Caisson). Multiple passages were used throughout these experiments for both cell lines with hTERT HFK passaging ranging from 15–80 and primary HFKs passaging ranging from 9–11. hTERT HFKs and primary HFKs both expressed the control vector (LXSN) and β-HPV 8E6; hTERT HFKs expressed HA-tagged β-HPV 8E6. In total, one primary HFK and one hTERT HFK cell line (each from separate donors) was used in these experiments.

### 4.2. Cell Cycle Analysis

Cells were harvested by trypsinization from 10-cm dishes, with cells being 70–90% confluent. After washing with cold 1× phosphate-buffered saline (PBS), cells were fixed with 4% paraformaldehyde (PFA) in 1×PBS for 15 min, and permeabilized in PBS containing 0.2% Triton X-100 for 30 min at room temperature. After washing with PBS, cells were resuspended in 0.2 mL of PBS and 3 μM of DAPI was added, then incubated at room temperature for 30 min in the dark [68].

Samples were analyzed by using an LSRFortessa X20 Flow Cytometer (BD, Franklin Lakes, NJ). Cells were gated on the Forward versus Side Scatter plot to eliminate debris, and then single cells were gated by using a dot-plot showing the pulse width versus pulse area of the DAPI channel. Post-acquisition analysis was performed with Flowing software 2.5.1. [68].

### 4.3. Comparative Transcriptomic Analysis

Web-based software on cBioPortal for Cancer Genomics (ww.cbioportal.org) was used to analyze RNAseq data from the *Cancer Cell Line Encyclopedia* [44,45,46]. List of genes for each category in Figure 1 and Appendix A is provided here: NER genes: UBE2B, FAAP20, POLK, PRIMPOL, RFC1, POLE3, RPA1, POLD1, RPA3, PCLAF, POLE2, RFC5, DTL, PCNA, RFC4, POLD3, RFC2, RPA2, ZBTB1, POLI, REV3L, REV1, POLH, VCP, RAD18, ISG15, SPRTN. TLS genes: CDK7, POLE, POLE2, POLE3, POLD1, POLD2, GTF2H1, GTF2H4, POLD3, POLD4, POLE4, RBX1, PCNA, CCNH, DDB2, ERCC8, DDB1, RPA3, LIG1, RFC1, RFC2, RFC3, RFC4, RFC5, XPC, ERCC6, MNAT1, ERCC3, ERCC2, GTF2H5, XPA, ERCC4, ERCC1. ATR/ATM genes: CHEK1, CDC25A, BRCA1, TP53. List of genes in Appendix A is provided here: BRCA1, MRE11, RAD9A, RAD9B, RAD50, TP53, NBN, PRKDC, RBBP8, ATMIN, HIF1A, TOPBP1, TP53BP1, MDC1, H2AFX, STRAP, SMC1B, E2F1, AATF, DCLRE1C, MDC1, EXO1, DNA2

### 4.4. Immunoblot

Once cell lines were 85% confluent after being seeded onto 6-well plates, they were exposed to 5 mJ/cm^2^ UV radiation for the appropriate time. Then, whole cell lysates were prepared by washing cells in cold 1×PBS before incubating on ice in complete RIPA lysis buffer (RIPA lysis buffer, protease inhibitor, phosphatase inhibitor) and mechanically harvested. Lysates were then centrifuged for higher purification and protein concentration was determined via BCA assay. 20 μg protein lysates were electrophoresed on SDS-PAGE and transferred to Immobilon-P membranes (Millipore, Burlington, MA, USA). The membranes were then probed with primary and secondary antibodies. All key immunoblot results were repeated at least five times (three times in hTERT HFKs and twice in HFKs to confirm the phenotype). Negative results (e.g., sub-cellular fractionation experiments) were done in duplicate. Quantification was performed using ImageJ (NIH, Rockville, MD, USA).

### 4.5. Antibodies

The following primary antibodies were used: pATM (Ser1981) (D25E5) (13050S, Cell Signaling, Danvers, MA), ATM (11G12) (92356S, Cell Signaling), pATR (Thr1989) (58014S, Cell Signaling), ATR (2790S, Cell Signaling), pBRCA1 (Ser1423) (ab90528, Abcam, Cambridge, United Kingdom), BRCA1 (9010S, Cell Signaling), pCHK2 (Thr68) (C13C1)( 2197S, Cell Signaling), CHK2 (2662S, Cell Signaling), pCHK1 (Ser345) (133D3) (2348S, Cell Signaling), CHK1 (2G1D5) (2360S, Cell Signaling), pCDC25A (Thr507) (PA512564, Thermo Fisher, Waltham, MA), CDC25A (DCS121) (MA112293, Thermo Fisher), pCDC2 (Thr14) (2543S, Cell Signaling), CDC2 (77055S, Cell Signaling), pCDK2 (Tyr15) (PA5-77907, Fisher Scientific, Hampton, NH), CDK2 (78B2) (2546S, Cell Signaling), pXPA (Ser196) (PA5-64730, Thermo Fisher), XPA (5F12) (ab65963, Abcam), RAD18 (ab57447, Abcam), UB. PCNA (Lys164) (D5C7P) (13439S, Cell Signaling), PCNA (PC10) (2586S, Cell Signaling), POLκ (ab57070, Abcam), POLη (B-7) (sc-17770, Santa Cruz, Dallas, TX), pRPA32/RPA2 (Ser8) (83745S, Cell Signaling), RPA32/RPA2 (52448S, Cell Signaling), RPA70/RPA1 (2267S, Cell Signaling), TOPBP1 (B-7) (sc-271043, Santa Cruz), GAPDH (0411) (sc-47724, Santa Cruz), Nucleolin (C23) (MS-3) (sc-803, Santa Cruz).

The following secondary antibodies were used: Peroxidase AffiniPure Goat Anti Mouse IgG (H + L) (115-035-003, Jackson ImmunoResearch, West Grove, PA), Anti Rabbit IgG, HRP-linked (7074S, Cell Signaling), Goat anti-Rabbit IgG (H + L) Cross-Adsorbed Secondary Antibody, Goat anti-Mouse IgG (H + L) Cross-Adsorbed Secondary Antibody, Alexa Fluor 488 (A-11001, Thermo Fisher, Waltham, MA), Alexa Fluor 594 (A-11012, Thermo Fisher).

### 4.6. Immunofluorescent Microscopy

Cells were seeded onto glass bottom plates (Cellvis, Mountain View, CA, USA), grown for 24 h and exposed to 5 mJ/cm^2^ UV radiation. Then once it was the appropriate time after 5 mJ/cm2 UV exposure, the cells were incubated in 4% formaldehyde for 15 min. Then the cells were permeabilized with 0.1% Triton X for 10 min. Next, the cells were blocked with 3% BSA and incubated with primary antibody overnight at 4 °C. The next day, the cells were incubated with fluorescent secondary antibodies (1:500) for 1 h and stained with 300 nM DAPI (D1306, Thermo Fisher) for 9 min. Cells were imaged using the Carl Zeiss 700 confocal microscope (Oberkochen, Germany) using the 40× (1.4 NA Oil) objective. Foci and intensity analyses were completed using ImageJ.

### 4.7. Subcellular Fractionation

Cells were seeded and grown for 24 h before being exposed to 5 mJ/cm^2^ UV and incubated for the appropriate time after radiation. Whole cell lysates were prepared by washing cells in cold 1×PBS before mechanically harvesting the cells in Subcellular Fractionation Buffer (HEPES, KCl, MgCl2, EDTA, EGA, pH 7.4, 1mM DTT, protease inhibitor, and phosphatase inhibitor). Nuclear and cytoplasmic lysates were separated through centrifugation. 20 μg protein lysates were electrophoresed on SDS-PAGE and transferred to Immobilon-P membranes (Millipore, Burlington, MA, USA). The membranes were then probed with primary and secondary antibodies.

### 4.8. mRNA Quantification

Cell were lysed using Trizol (Invitrogen, Carlsbad, CA, USA) and RNA isolated with the RNeasy kit (Qiagen, Hilden, Germany). Two μg of RNA were reverse transcribed using the iScript™ cDNA Synthesis Kit (Bio-Rad, Hercules, CA). Quantitative reverse transcription-PCR was performed in triplicate with the TaqMan™ FAM-MGB Gene Expression Assay (Applied Biosystems, Foster City, CA) and C1000 Touch Thermal Cycler (Bio-Rad). The following probes (Thermo Scientific) were used: ACTB (Hs01060665_g1), POLH (Hs00197814_m1), POLK (Hs00211965_m1), CHEK1 (Hs00967506_m1), XPA (Hs00166045_m1)

### 4.9. UV Radiation

Cells were washed with 1×PBS and then irradiated at 5 mJ/cm^2^ using the UV Stratalinker 2400 (Stratagene, San Diego, CA, USA). Then media was added back to the cells and they were allowed to incubate for the appropriate time after UV exposure.

### 4.10. Statistical Analysis

Statistical significance was determined using student’s *t*-test. p values less than or equal to 0.05 were reported as significant.

## Figures and Tables

**Figure 1 pathogens-08-00267-f001:**
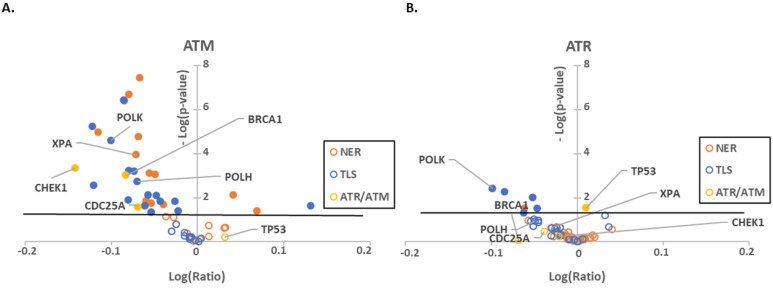
Low expression of ATR/ATM mRNA correlates with a decrease in UV damage repair pathways gene expression. Volcano plots comparing RNAseq data of NER (orange), TLS (blue) and ATR/ATM target (yellow) genes between cell lines (**A**) with low ATM expression (z-score > 2) and without decreased ATM expression (z-score < 2) or (**B**) between cells with (z-score > 2) and without (z-score < 2) low ATR expression. Outlined circles represent non-significant expression changes. Filled in circles represent significant expression changes. The black line represents significance cutoff (*p* < 0.05). The x-axis depicts the log of the ratio of each gene’s expression levels in cell lines with high expression of ATM/ATR versus all other cell lines in the cancer cell line encyclopedia. The y-axis shows the negative log of the *p*-value. Genes with reduced expression appear to the left of the y-axis, while genes with increased expression are on the right.

**Figure 2 pathogens-08-00267-f002:**
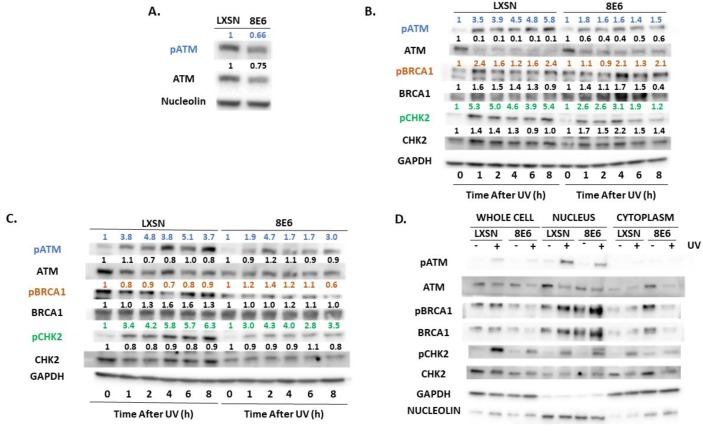
β-HPV 8E6 attenuates ATM activation. (**A**) Representative immunoblots of untreated hTERT HFKs with vector control (LXSN) and β-HPV 8E6 cell lines. Nucleolin was used as a loading control. (**B**) Representative immunoblots of hTERT HFKs with vector control (LXSN) and β-HPV 8E6 harvested 0–8 h post 5 mJ/cm^2^ UVR. GAPDH was used as a loading control. (**C**) Representative immunoblots of primary HFKs with vector control (LXSN) and β-HPV 8E6 harvested 0–8 h post 5 mJ/cm^2^ UVR. GAPDH was used as a loading control. (**A**–**C**) The numbers above bands represent quantification by densitometry. This is shown relative to untreated cells within the same cell line and normalized to the loading control. (**D**) Subcellular fractionation of hTERT HFKs with vector control (LXSN) and β-HPV 8E6 cell line lysates harvested 6 h post exposure to 5 mJ/cm^2^ UVR were observed via immunoblot. GAPDH was used as a cytoplasmic loading control and Nucleolin was used as a nuclear loading control.

**Figure 3 pathogens-08-00267-f003:**
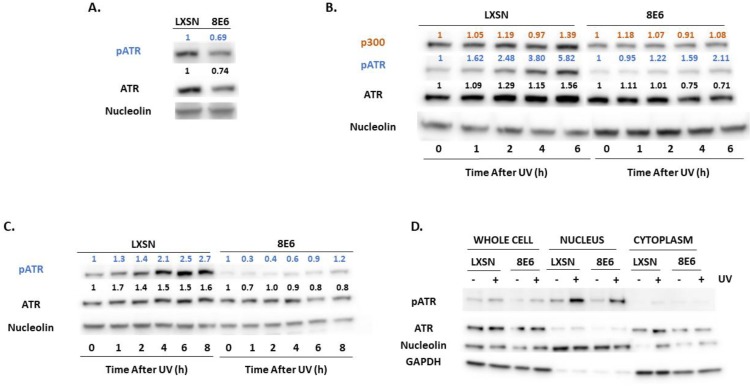
β-HPV 8E6 attenuates ATR activation. (**A**) Representative immunoblots of untreated hTERT HFKs with vector control (LXSN) and β-HPV 8E6 cell lines. Nucleolin was used as a loading control. (**B**) Representative immunoblots of hTERT HFKs with vector control (LXSN) and β-HPV 8E6 harvested 0–6 h post 5 mJ/cm^2^ UVR. Nucleolin was used as a loading control. (**C**) Representative immunoblots of primary HFKs with vector control (LXSN) and β-HPV 8E6 harvested 0–8 h post 5 mJ/cm^2^ UVR. Nucleolin was used as a loading control. (**A**–**C**) The numbers above bands represent quantification by densitometry. This is shown relative to untreated cells within the same cell line and normalized to the loading control. (**D**) Subcellular fractionation of hTERT HFKs with vector control (LXSN) and β-HPV 8E6 cell line lysates harvested 6 h post exposure to 5 mJ/cm^2^ UVR were observed via immunoblot. GAPDH was used as a cytoplasmic loading control and Nucleolin was used as a nuclear loading control.

**Figure 4 pathogens-08-00267-f004:**
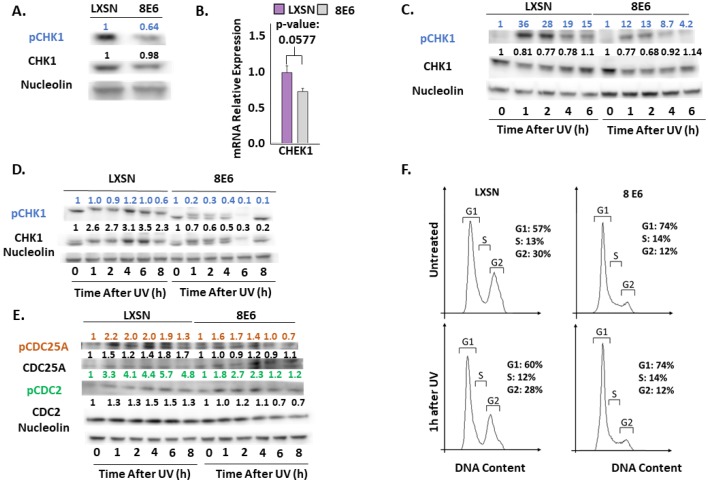
β-HPV 8E6 attenuates CHK1 phosphorylation. (**A**) Representative immunoblots of untreated hTERT HFKs with vector control (LXSN) and β-HPV 8E6 cell lines. Nucleolin was used as a loading control. (**B**) mRNA expression level of CHEK1 in vector control (LXSN) and β-HPV 8E6 expressing primary HFKs as measured by RT-qPCR and normalized towards the expression level of β-actin. Data shown in figures are the means of ±SE of three independent experiments. (**C**) Representative immunoblots of hTERT HFKs with vector control (LXSN) and β-HPV 8E6 harvested 0–6 h post 5mJ/cm^2^ UVR. Nucleolin was used as a loading control. (**D**) Representative immunoblots of primary HFKs with vector control (LXSN) and β-HPV 8E6 harvested 0–8 h post 5 mJ/cm^2^ UVR. Nucleolin was used as a loading control. (**E**) Representative immunoblots of hTERT HFKs with vector control (LXSN) and β-HPV 8E6 harvested 0–8 h post 5 mJ/cm^2^ UVR. Nucleolin was used as a loading control. (**A, C**–**E**) The numbers above bands represent quantification by densitometry. This is shown relative to untreated cells within the same cell line and normalized to the loading control. (**F**) Cell cycle analysis of hTERT HFKs with LXSN vector control and β-HPV 8E6 1 h post 5 mJ/cm^2^ UVR.

**Figure 5 pathogens-08-00267-f005:**
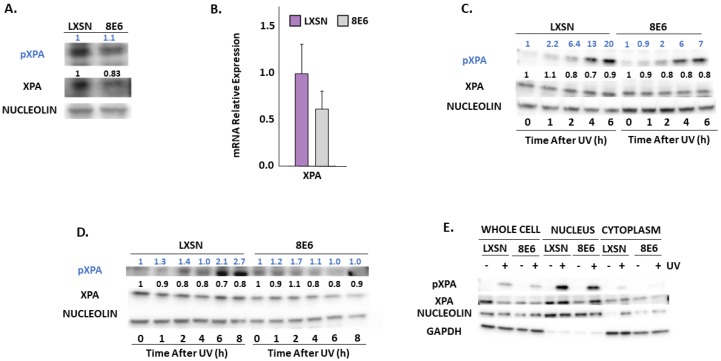
β-HPV 8E6 attenuates XPA phosphorylation. (**A**) Representative immunoblots of untreated hTERT HFKs with vector control (LXSN) and β-HPV 8E6 cell lines. Nucleolin was used as a loading control. (**B**) mRNA expression level of XPA in vector control (LXSN) and β-HPV 8E6 expressing primary HFKs as measured by RT-qPCR and normalized towards the expression level of β-actin. Data shown in figures are the means of ±SE of three independent experiments. (**C**) Representative immunoblots of hTERT HFKs with vector control (LXSN) and β-HPV 8E6 harvested 0–6 h post 5 mJ/cm^2^ UVR. Nucleolin was used as a loading control. (**D**) Representative immunoblots of primary HFKs with vector control (LXSN) and β-HPV 8E6 harvested 0–8 h post 5 mJ/cm^2^ UVR. Nucleolin was used as a loading control. (**A**,**C**,**D**) The numbers above bands represent quantification by densitometry. This is shown relative to untreated cells within the same cell line and normalized to the loading control. (**E**) Subcellular fractionation of hTERT HFKs with vector control (LXSN) and β-HPV 8E6 cell line lysates harvested 6 h post exposure to 5 mJ/cm^2^ UVR were observed via immunoblot. GAPDH was used as a cytoplasmic loading control and Nucleolin was used as a nuclear loading control.

**Figure 6 pathogens-08-00267-f006:**
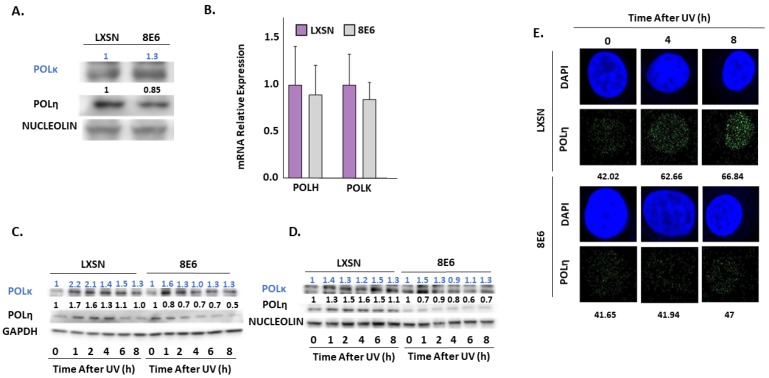
β-HPV 8E6 attenuates POLη abundance. (**A**) Representative immunoblots of untreated hTERT HFKs vector control (LXSN) and β-HPV 8E6 cell lines. Nucleolin was used as a loading control. (**B**) mRNA expression level of POLH and POLK in vector control (LXSN) and β-HPV 8E6 expressing primary HFKs as measured by RT-qPCR and normalized towards the expression level of β-actin. Data shown in figures are the means of ±SE of three independent experiments. (**C**) Representative immunoblots of hTERT HFKs with vector control (LXSN) and β-HPV 8E6 harvested 0–8 h post 5 mJ/cm^2^ UVR. GAPDH was used as a loading control. (**D**) Representative immunoblots of primary HFKs with vector control (LXSN) and β-HPV 8E6 harvested 0–8 h post 5 mJ/cm^2^ UVR. Nucleolin was used as a loading control. (**A**,**C**,**D**) The numbers above bands represent quantification by densitometry. This is shown relative to untreated cells within the same cell line and normalized to the loading control. (**E**) Representative immunofluorescence microscopy images of hTERT HFKs. POLη (green) and nuclei stained (blue) with DAPI.

**Figure 7 pathogens-08-00267-f007:**
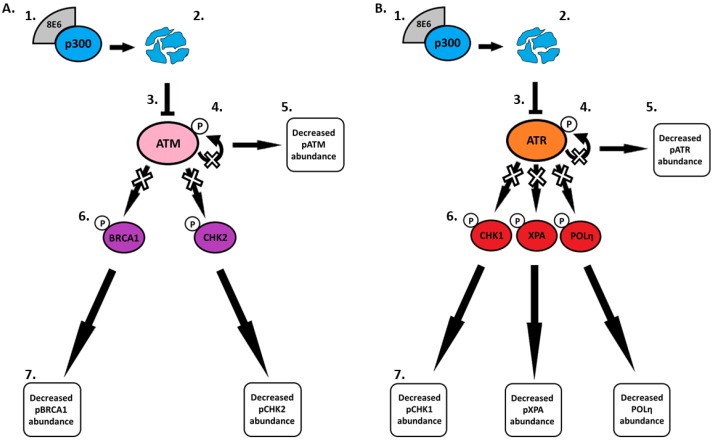
Schematic diagram of the effects of β-HPV 8E6 on downstream ATM and ATR targets. (**A**) β-HPV 8E6 binds to p300 (1) causing p300 to become destabilized and subsequentially degraded (2). The decrease in p300 levels leads to less ATM transcription (3). This leads to a decrease in ATM autophosphorylation (4) resulting in less activated ATM available (5). Limited availability of activated ATM leads to a decrease in ATM-dependent phosphorylation of downstream proteins (6) causing changes in β-HPV 8E6 infected cells (7). (**B**) β-HPV 8E6 binds to p300 (1) causing p300 to become destabilized and subsequentially degraded (2). The decrease in p300 levels leads to less ATR transcription (3). This leads to a decrease in ATR autophosphorylation (4) resulting in less activated ATR available (5). Limited availability of activated ATR leads to a decrease in ATR-dependent phosphorylation of downstream proteins (6) causing changes in β-HPV 8E6 infected cells (7).

**Table 1 pathogens-08-00267-t001:** Different expression profiles in cells with lower ATM, ATR and p53 expression. n.s. denotes a non-significant relationship. −/+ denote significant relationships *p* < 0.05 with low magnitude. −−/++ denote relationships with 0.05 < *p* > 0.001 and 0.02 > log ration > 0.01. −−−/+++ denote relationships with *p* < 0.001 and log ratio > 0.02. (sign denotes negative and positive regulation).

	p53	ATR	ATM
BRCA1	n.s.	++	+++
CDC25A	n.s.	n.s.	++
CHEK1	−	n.s.	+++
POLH	+++	n.s.	++
POLK	+	++	+++
XPA	−	n.s.	+++

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
