# Peer review of "β-HPV 8E6 Attenuates ATM and ATR Signaling in Response to UV Damage"

_pathogens, 2019, doi:10.3390/pathogens8040267_

Round 1
Reviewer 1 Report
Dr Wallace’s group has shown in the past that betaHPV E6 targets ATM and ATR, two essential kinases involved in the DNA damage response. In the current manuscript they aimed at describing the down-stream effects of modulating ATM and ATR. They show that E6 reduces ATR-dependent phosphorylation required for XPA stabilization and NER function. In addition, decreased ATM activation also led to inability to phosphorylate downstream targets involved in the repair of DNA double strand breaks. Before publication the authors need to address the following points:
General comments:
The authors need to include some Western blot quantifications throughout the manuscript as some changes are not strong. In addition, the supplementary figures need better legends to fully understand the experiments / results presented.
Specific comments:
Line 32: replace „oral cavity“ with „oropharynx“ and respective references. Line75: please indicate from which cell line/type the transcriptomic data was collected from. Legend to figure 1: Line 92: change ATR to ATM Line 98: change ATM to ATR Combine legend to a) with b): it is exactly the same text Include a bit more description to understand the x-axis Line 114: delete“ does not likely occur“ and change to „... ATM/ATR is not exclusively mediated through p53“ Line 116: -- does not exist, please delete Line 117: --- does not exist, please delete Figure 2D: Although the authors state in lines 143 and 144, that there was no robust differences in localization of the proteins analysed, a difference, however, can be seen in the cytoplasm fraction. Here, in untreated cells, E6 seems to hold the proteins back in the cytoplasm when compared to the control lanes. Lines 235-237: These successive sentences bring exactly opposite statements for the same result. Please correct. In addition, what exactly should be seen in suppl. Figure 6B, C? I do not see major differences. 6: The green signals can be stronger. It is too faint in its current form. Discussion: The discussion is not focused enough. Delete the sections with CRISPR/CAS9 and hTERT and focus on the newly gathered results. Be more focused.
Author Response
Reviewer#1
Dr Wallace’s group has shown in the past that betaHPV E6 targets ATM and ATR, two essential kinases involved in the DNA damage response. In the current manuscript they aimed at describing the down-stream effects of modulating ATM and ATR. They show that E6 reduces ATR-dependent phosphorylation required for XPA stabilization and NER function. In addition, decreased ATM activation also led to inability to phosphorylate downstream targets involved in the repair of DNA double strand breaks. Before publication the authors need to address the following points:
The authors need to include some Western blot quantifications throughout the manuscript as some changes are not strong.
Thank you for the excellent suggestion, we have added Western blot quantifications to the western blot figures in order to highlight the changes. We also provide graphical representations of densitometry as a supplemental figure (Supplementary Figure 4). The one exception to this was that we felt uneasy presenting quantified protein values for the subcellular fractionation experiments. This would require normalizations to different loading controls for proteins on the same blot making us worry that the data would have been over-manipulated. Because we left out these values, we tempered our interpretations of the fractionation analysis.
The supplementary figures need better legends to fully understand the experiments / results presented.
We have added legends for the supplementary figures.
Line 32: replace „oral cavity“ with „oropharynx“ and respective references.
Done
Line75: please indicate from which cell line/type the transcriptomic data was collected from.
In our original submission, we referenced the cancer cell line encyclopedia, the original source of the interrogated transcriptomic data. This database contains RNAseq data from 877 different cell lines, making it a robust resource for this type of computational analysis. Unfortunately, the size of the database functionally limits our ability to provide an exhaustive list of the cell lines with and without changes in our genes of interest. Instead, we provide the number of cell lines that were compared for each graph in Figure 1 and Supplemental Figure 1 (lines 71-72, and 82). As an alternative, we could also provide the names of the cell lines with altered expression in a supplemental figure. As we are certain that readers will have this same question, we thank you for this suggestion.
Legend to figure 1: Line 92: change ATR to ATM Line 98: change ATM to ATR
ATM and ATR has been changed. Thank you for pointing out this error.
Combine legend to a) with b): it is exactly the same text Include a bit more description to understand the x-axis
We have combined the description of the graph in Figure 1A and B in order to decrease the repetitiveness of the figure legend. We have also added a more thorough description to the figure legend to describe the X-axis (Lines 90-92).
Line 114: delete“ does not likely occur“ and change to „... ATM/ATR is not exclusively mediated through p53“
Done
Line 116: -- does not exist, please delete Line 117: --- does not exist, please delete
Although we are not clear on the meaning of this comment, we have clarified our language in this section of the manuscript in hopes of addressing the reviewer’s remark.
Figure 2D: Although the authors state in lines 143 and 144, that there was no robust differences in localization of the proteins analysed, a difference, however, can be seen in the cytoplasm fraction. Here, in untreated cells, E6 seems to hold the proteins back in the cytoplasm when compared to the control lanes.
We have added a statement about the cytoplasmic localization results that we observed in Figure 2D. Thank you for the suggestion, you can see the statement on lines 142-144.
Lines 235-237: These successive sentences bring exactly opposite statements for the same result. Please correct.
Thank you for pointing this out, we have changed the wording in order to better explain our results (lines 245-249). We meant to point out that while POLH (mRNA) was not decreased, POLη (protein) was. This is consistent with the known requirement of ATR phosphorylation to stabilize POLη (protein).
In addition, what exactly should be seen in suppl. Figure 6B, C? I do not see major differences.
For supplemental Figure 6B and C, we wanted to highlight the differences in XPA localization between vector control (LXSN) and β-HPV 8E6. XPA is almost exclusively nuclear in control cells after UV but is distributed evenly throughout the cell when β-HPV 8E6 is expressed. We are sorry that this was not clear, we have reworded the description in the paper in order to correct our mistake (lines 231-235).
6: The green signals can be stronger. It is too faint in its current form.
We apologize for the faint signal in the IF images, changes have been made to increase the strength of POLη (green signal) in figure 6E so that it is easier to interpret.
Discussion: The discussion is not focused enough. Delete the sections with CRISPR/CAS9 and hTERT and focus on the newly gathered results. Be more focused.
We appreciate the feedback and have deleted the sections with CRISPR/Cas9 and hTERT in order to focus more on the implications and restrictions of our results. The new discussion is more focused and addresses some of reviewer #2’s comments.
Reviewer 2 Report
Summary:
Snow et al show that expression of β-HPV E6 in cell culture models causes a lowering of ATM and ATR levels, that this appears to lead to decreased activity of downstream effectors, and reduced DNA repair response following UV radiation. The study is well written, and the authors have drawn reasonable conclusions based on the data. I have noted quite a list of revisions, though.
Major:
1.
The in silico analysis of gene expression in cell lines is not well described and over-interpreted. It is unclear to me what is gained by performing this analysis, as the choice of subsequent analyses seems completely decoupled from the results in this section. I would consider removing this section from the manuscript as it stands now, as I can see little merit to the section and no loss to the study as a whole if it were to be removed. If it is to be retained, some minimum requirements must be met: The authors must refer to the observations in this section exclusively as correlational, and not use causal language; the potential effect of ATM/ATR/p53 on DNA repair gene expression cannot be inferred from this kind of data. The cell lines on which the data are based must be listed. The genes defined to be in each category of DNA repair must be listed. Some optional amendments: Consider stratifying the expression by mutation status of ATM/ATR/p53. Consider restricting the analysis to cell lines relevant to the current study (keratinocytes, for example). Refer to low expression as “low expression”, rather than “under-expressed”. Use log10 for p-values and log2 for ratio in volcano plot (more easily readable). Report p-values corrected for multiple comparisons.
As an indication of the level of HPV E6 expression, the authors should have measured baseline mRNA expression in each of the experimental settings.
3.
The discussion lacks important sections that will help the reader assess the reliability of the findings:
The limitations of the study need to be discussed To what extent the model system is biologically relevant needs to be discussed. Whether the findings are in line with previous reports needs to be discussed.
Minor:
4.
In the abstract, the authors state that “β-HPV E6’s hindrance of ATM/ATR signaling during UV-associated cell cycle 17 arrest was incomplete”. That the negative effect of β-HPV E6 on ATM/ATR signalling was incomplete is likely true, both for shifts in cell cycle distribution and other phenotypic traits described. The incompleteness statement could be more broadly applied.
5.
In the abstract, there could be a summary "interpretation" statement along the lines presented in figure 7, where the observations are interpreted as likely being caused by downregulation of ATM/ATR by β-HPV E6, rather than the statement that the findings are consistent with a role in NMSC.
6.
The introduction is well written and presents a good rationale for conducting the study. The last paragraph though, should focus less on listing the results, and more on describing the approach.
7.
The authors argue that the observations in this study are consistent with downregulation of ATM/ATR expression. To verify this, mRNA levels of ATM/ATR could have been determined using qPCR.
8.
The study relies heavily on small differences in immunoblot intensities. These differences are hard to discern, and could have benefitted from densitometry measures. In particular so that the levels of one protein, or protein modification, could be corrected for the loading control and other dependency relations.
9.
In figure legends, it is stated that the immunoblots are representative. How many parallel experiments were run?
10.
In figure legends, time points for harvest of cells is described as 1–6/8 hours after UV, when it says 0…6/8 in the figure.
11.
In figure 4f there is a substantial decrease in G2/M associated with β-HPV E6 expression. Can the authors comment on this? I am not sure I agree that the changes in cell cycle distribution are “subtle”. In addition, the thresholds used for each stage of the cell cycle should be indicated.
12.
In section 4.2, multiple passages are said to have been used. It would be more interesting to know the range of passages used.
13.
If the in silico analysis is used, it needs to be described better (section 4.4).
14.
The description of the immunoblot is confusing. Were the cells exposed to UV after lysis?
15.
The section on antibodies might be moved to after the Immunoblot section.
16.
The UV radiation should be described in more detail.
17.
More details on cell culturing prior to harvesting for immunoblots are needed. How many cells were seeded, for how long were they cultured, at which confluence were they harvested, etc..?
Author Response
Reviewer#2
Snow et al show that expression of β-HPV E6 in cell culture models causes a lowering of ATM and ATR levels, that this appears to lead to decreased activity of downstream effectors, and reduced DNA repair response following UV radiation. The study is well written, and the authors have drawn reasonable conclusions based on the data. I have noted quite a list of revisions, though.
Major:
1.
The in silico analysis of gene expression in cell lines is not well described and over-interpreted. It is unclear to me what is gained by performing this analysis, as the choice of subsequent analyses seems completely decoupled from the results in this section. I would consider removing this section from the manuscript as it stands now, as I can see little merit to the section and no loss to the study as a whole if it were to be removed. If it is to be retained, some minimum requirements must be met: The authors must refer to the observations in this section exclusively as correlational, and not use causal language; the potential effect of ATM/ATR/p53 on DNA repair gene expression cannot be inferred from this kind of data.
We changed the language to stress that these are correlational changes and not causational. Thank you for pointing out this overstatement in our description of the in silico data. With the reviewer’s suggestions, we believe the data is now better described and interpreted. (Lines 80-88)
The cell lines on which the data are based must be listed.
This suggestion has been addressed above in the Reviewer #1 comments number 4. Briefly, transcriptomic data from over 1000 cell lines were used in this analysis. As a result, it would be untenable to list each cell line. Our hope is that changes address the reviewer’s comment. (lines 71-72, and 82)
The genes defined to be in each category of DNA repair must be listed.
We have added the list of genes that was used for each category to the methods section titled “Comparative Transcriptomic Analysis” (lines 351-360). These genes were chosen based on their KEGG or MG1 Gene Ontology designation.
Some optional amendments: Consider stratifying the expression by mutation status of ATM/ATR/p53.
We have added a supplementary figure to show ATM/ATR by mutation status (lines 83-85).
Refer to low expression as “low expression”, rather than “under-expressed”.
We have changed the wording for the in silico data to refer to low expression as “low expression” rather than “under-expressed” (Lines 89-97).
Use log10 for p-values and log2 for ratio in volcano plot (more easily readable). Report p-values corrected for multiple comparisons.
We have changed the volcano plots so that the graph has log10 for p-values and log 2 for the ratios, as suggested.
As an indication of the level of HPV E6 expression, the authors should have measured baseline mRNA expression in each of the experimental settings.
We have confirmed HPV E6 expression both functionally (p300 levels) and directly by immunoblot against HA. The cell lines have also been used characterized as part of other studies. Because there is no reason to expect that expression to change between individual experiments, we only confirmed expression of HPV E6 once. Since this approach prevented us precisely correlating β-HPV 8E6 expression with ATR/ATM signaling, we have been careful to avoid these claims. Instead, we only state that β-HPV 8E6 has the ability to hinder ATM/ATR signaling after UV.
The discussion lacks important sections that will help the reader assess the reliability of the findings: The limitations of the study need to be discussed To what extent the model system is biologically relevant needs to be discussed. Whether the findings are in line with previous reports needs to be discussed.
Thank you for these suggestions. They have been incorporated into the discussion. Please see our comments to Reviewer #1’s point #13 for more details.
Minor:
In the abstract, the authors state that “β-HPV E6’s hindrance of ATM/ATR signaling during UV-associated cell cycle 17 arrest was incomplete”. That the negative effect of β-HPV E6 on ATM/ATR signalling was incomplete is likely true, both for shifts in cell cycle distribution andother phenotypic traits described. The incompleteness statement could be more broadly applied.
We have reworded the abstract (lines 19-20).
In the abstract, there could be a summary "interpretation" statement along the lines presented in figure 7, where the observations are interpreted as likely being caused by downregulation of ATM/ATR by β-HPV E6, rather than the statement that the findings are consistent with a role in NMSC.
We have deleted the statement about NMSC in order to include a summary interpretation statement (lines 18-20).
The introduction is well written and presents a good rationale for conducting the study. The last paragraph though, should focus less on listing the results, and more on describing the approach.
We changed the last paragraph of the introduction to focus on our approaches rather than summarizing the results (lines 51-64).
The authors argue that the observations in this study are consistent with downregulation of ATM/ATR expression. To verify this, mRNA levels of ATM/ATR could have been determined using qPCR.
We agree with the reviewer that it is important to demonstrate the β-HPV E6 acts by inhibiting expression. However, we have previously published that ATR mRNA is decreased by qPCR, limiting our ability to reproduce that in this manuscript (Wallace et al 2012). For ATM, we now avoid directly stating that β-HPV 8E6 acts at the transcript level and instead just discuss the decreased protein abundance demonstrated in Figure 2.
The study relies heavily on small differences in immunoblot intensities. These differences are hard to discern, and could have benefitted from densitometry measures. In particular so that the levels of one protein, or protein modification, could be corrected for the loading control and other dependency relations.
Thank you for the suggestion, as noted above (Reviewer #1 point 1), we now include quantifications in each blot.
In figure legends, it is stated that the immunoblots are representative. How many parallel experiments were run?
All key immunoblot results were repeated at least five times (three times in hTERT HFKs and twice in HFKs to confirm the phenotype). Negative results (e.g. sub-cellular fractionation experiments) were done in duplicate. The strength of our conclusions parallels these differences. We have added this information to our resubmission (Lines 369-372)
In figure legends, time points for harvest of cells is described as 1–6/8 hours after UV, when it says 0…6/8 in the figure.
We apologize for the inconsistency and have corrected the figure legends to say 0…6/8. (Lines 148-151)
In figure 4f there is a substantial decrease in G2/M associated with β-HPV E6 expression. Can the authors comment on this? I am not sure I agree that the changes in cell cycle distribution are “subtle”. In addition, the thresholds used for each stage of the cell cycle should be indicated.
Unfortunately, the decrease in G2/M was not consistent between experiments. While it may be real, the only consistency we found in all three repeats was the change in the G1 population. Because of the variability, we chose the word “subtle” to reflect our uncertainty. We have added language to this effect in the resubmission. (Lines 290-293)
In section 4.2, multiple passages are said to have been used. It would be more interesting to know the range of passages used.
We have added the range of passages as suggested which you can see in the M&M (lines 333-334), thank you for the suggestion.
If the in silico analysis is used, it needs to be described better (section 4.4).
We have improved the description of our in silico analysis (lines 350-360).
The description of the immunoblot is confusing. Were the cells exposed to UV after lysis?
We have addressed the lack of clarity in our resubmission (lines 363-366).
The section on antibodies might be moved to after the Immunoblot section.
Done
The UV radiation should be described in more detail.
We added more detail to UV radiation (lines 422-424).
More details on cell culturing prior to harvesting for immunoblots are needed. How many cells were seeded, for how long were they cultured, at which confluence were they harvested, etc..?
We added more information about the cell culturing prior to harvesting for immunoblots, most notably by including the confluency of the cell lines before they were harvested for immunoblots (lines 363).
Reviewer 3 Report
The manuscript by Snow et al focuses on beta HPVs, which are associated with NMSCC, and their disruption of UV associated DNA Damage Repair pathways. This group leverages both publicly available gene expression data (through cBioPortal) to identify key target proteins in the DNA repair pathways downstream of ATM and ATR and then uses the laboratory's own tissue culture studies to validate these targets in UV exposure assays. Normally, DDR pathways proteins would be activated by UV exposure to drive DDR, and yet they are inadequately activated when beta HPV E6 is expressed in these cells. Overall the data is clear and support the hypothesis of the project.
There are two minor comments for the authors. First, is unclear how many cell lines (HFK cell lines and hTERT HFK cell lines) were used in generating the data for the manuscript. The M&M do not describe the number of cell lines used, and the figure legends also allude to data being representative of the experiments, but not the number of experiments executed. It is also unclear if the experiments are repeated within one donor cell line (one HFK and one hTERT HFK), or if different donor cell lines are used in one or more experimental protocol. That should be clarified in the M&M and the figure legends. Second, the data from the western blots may be strengthened by quantification of the band intensity using a software, such at ImageJ. Westerns are not fully quantitative, but there are some loading differences across LXSN versus 8E6 lysates that would be adjusted for with relative quantification. Making quantification relative to a loading control (and also relative to the LXSN lysate specifically at time 0) would help emphasize your results during the UV experiments.
Simple copy editing notes:
Line 134 has a space before the comma
Line 196 has a font change for B-HPV
Line 283 has a space before the period
Author Response
Reviewer#3
The manuscript by Snow et al focuses on beta HPVs, which are associated with NMSCC, and their disruption of UV associated DNA Damage Repair pathways. This group leverages both publicly available gene expression data (through cBioPortal) to identify key target proteins in the DNA repair pathways downstream of ATM and ATR and then uses the laboratory's own tissue culture studies to validate these targets in UV exposure assays. Normally, DDR pathways proteins would be activated by UV exposure to drive DDR, and yet they are inadequately activated when beta HPV E6 is expressed in these cells. Overall the data is clear and support the hypothesis of the project.
There are two minor comments for the authors.
First, is unclear how many cell lines (HFK cell lines and hTERT HFK cell lines) were used in generating the data for the manuscript. The M&M do not describe the number of cell lines used
We added a statement about the number of cell lines used in this paper to the methods and materials section (lines 335-336).
and the figure legends also allude to data being representative of the experiments, but not the number of experiments executed. It is also unclear if the experiments are repeated within one donor cell line (one HFK and one hTERT HFK), or if different donor cell lines are used in one or more experimental protocol. That should be clarified in the M&M and the figure legends.
We added a statement about the number of cell lines used to the methods and materials section (Lines 369-372).
The data from the western blots may be strengthened by quantification of the band intensity using a software, such at ImageJ. Westerns are not fully quantitative, but there are some loading differences across LXSN versus β-HPV 8E6 lysates that would be adjusted for with relative quantification. Making quantification relative to a loading control (and also relative to the LXSN lysate specifically at time 0) would help emphasize your results during the UV experiments.
Thank you for this advice. These changes have been made and are addressed in response to Reviewer #1’s first comment.
Line 134 has a space before the comma
Done
Line 196 has a font change for B-HPV
Done
Line 283 has a space before the period
Done
Round 2
Reviewer 1 Report
As far as the answers to my questions are concerned, I am satisfied with the revision of the manuscript and have no further criticism.
Author Response
We thank the reviewer for their time and effort spend improving our work.
Reviewer 2 Report
The manuscript has been improved in some respects, but the inclusion of intensity measures has been done in such a way as to confuse the reader. Immunoblotting, the technique on which this study is based, is a semi-quantitative technique. Some variation in intensity measures is therefore expected. The reported values seem to be quite variable, and only small differences are seen in most cases. In some cases, the results from one blot is reported, when another shows results contradicting the authors’ conclusion. Without more robust measures it seems to me that the data presented is insufficient for the conclusions that are drawn.
Here are my comments for the revised manuscript:
The authors have presented intensity measures for immunoblots. However, when a different baseline is set for each antibody in each cell line, and there is no correction for loading (as far as I can see), the numbers presented are not easily interpretable. In figure 2b, for example, the intensity of ATM and phosphorylated ATM seems to be higher in 8E6-cells prior to UV than in controls (in contradiction to the conclusions of the study), but since both are presented as having an intensity of 1, this is hard to assess. ATM and p-ATM in figure 2: Why does ATM levels decrease in both LXSN and 8E6 following UV in hTERT HFKs (b), but not in primary HFKs? On lines 130–131, the levels of ATM and pATM are said to be lower in 8E6, but this is not consistent with the blot in 2b. In figure 2b and c, levels of pBRCA1 are shown. These are not consistent between experimental systems, and only the increase in hTERT cells is described in the text (lines 136–138). On lines 157–185 it is stated that pATR levels did not increase in 8E6 cells. In figure 3b, they do, while in 3c they do not (though there seems to be a difference in the intensity reported and the bands on the blot). The authors could have concluded that the increase was smaller, but not that there was no increase in pATR. The authors state on lines 175–176 that there is a decrease in total Chk1. In figure 4, the levels of Chk1 are consistently shown to be stable. The levels of CDC25A and CDC2 in figure 4E do not appear to show clear patterns of variation, are the authors sure that these findings are reproducible? The gating used for cell cycle analysis varies between experimental settings; in particular, the number of cells found to be in the S phase seems clearly related to the size of the S gate. In LXSN cells, delineating cells in the S phase seems very challenging, as the transition from G1 to G2/M is completely smooth. Drawing conclusions based on the amount of S phase cells here seems inappropriate. In addition, the differences between baseline LXSN and 8E6 cells seems large enough that the subtle differences in protein levels described in the study might be influenced by the state of the cells rather than by direct signalling through 8E6. While the direction of causality here could probably be in either direction (cell cycle changes -> ATM/ATR changes, or ATM/ATR changes -> cell cycle changes), the fact that the clear difference in abundance of G2 cells was not reproducible seems to suggest that the experimental system is not stable enough to make any clear inferences about the subtle differences the authors describe. On lines 222–223, the levels of pXPA and total XPA are said to be lowered by 8E6. For total XPA, the difference is very subtle. For pXPA, there does not seem to be any difference in C or D, and maybe a difference in A. This suggests that the subtle differences seen are not reproducible. There is also a difference between hTERT HFKs and primary HFKs (4–6 hours after UV). On line 246, polη is describes as being lower in 8E6-expressing cells. In figure 6c, the opposite is shown, suggesting that the result is not reproducible. The authors have conducted an in silico analysis, but this is not mentioned in the discussion. Why was the in silico analysis performed? And what, exactly, do the authors conclude based on the in silico analysis? The analysis of expression dichotomized by mutations in ATM/ATR (suppl fig 3). How do these results “support the role of these kinases as broad regulators of UV responsive gene expression”? A lack of significant associations could be interpreted as evidence that expression of these genes are independent of ATM/ATR activity. On line 102–104, the authors ask how much the association between ATM/ATR and other genes is due to p53. If they had performed the expression correlation separately in p53-mutated cell lines and p53-wt cell lines, any genes significant in both would presumably not be “regulated” by ATM/ATR through p53. It looks like something like this was the intention of the analysis in table 1, but what has been done here seems to be three separate differential expression analyses. If this is the case, determining whether an association is due to one factor, rather than another is not really that meaningful for finding out what regulates what. Supplemental figure 4 does not have high enough resolution, and seems to describe pATM in HFKs twice. I can understand that replicate immunoblots are not included in the main text, and even as supplementary information, but given the subtle differences and sometimes inconsistent results presented, including the replicates in the densitometry plots might improve the impression of the reliability of the findings.
Author Response
The manuscript has been improved in some respects, but the inclusion of intensity measures has been done in such a way as to confuse the reader. Immunoblotting, the technique on which this study is based, is a semi-quantitative technique. Some variation in intensity measures is therefore expected. The reported values seem to be quite variable, and only small differences are seen in most cases. In some cases, the results from one blot is reported, when another shows results contradicting the authors’ conclusion. Without more robust measures it seems to me that the data presented is insufficient for the conclusions that are drawn.
We certainly were not trying to confuse or deceive anyone and regret that our efforts gave the reviewer this impression. The quantification of the immunoblots was done in two ways and both were provided in our last submission, but the blurriness of the fourth supplemental figure may have obscured the second normalization. We have also added clarifying statement and caveats to reflect places where uncertainty or inconsistencies exist.
Here are my comments for the revised manuscript:
The authors have presented intensity measures for immunoblots. However, when a different baseline is set for each antibody in each cell line, and there is no correction for loading (as far as I can see), the numbers presented are not easily interpretable.
As mentioned above, densitometry data was normalized and presented in two manners. In the main text, we show data normalized to the loading control and then normalized to untreated cells within each cell line. This approach was presented in the primary figures because we wanted to highlight altered responses to UV stimulus. We also provide the data normalized to the loading control and then with both cell lines normalized to untreated vector control (see supplemental figure 4). This is provided as averaged data across each experimental repeat. In general, these analyses are complimentary. When they were not, we now mention and discuss the differences.
In figure 2b, for example, the intensity of ATM and phosphorylated ATM seems to be higher in 8E6-cells prior to UV than in controls (in contradiction to the conclusions of the study), but since both are presented as having an intensity of 1, this is hard to assess.
Please see our response above and also lines 131-132 for changes more specific to this particular example.
ATM and p-ATM in figure 2: Why does ATM levels decrease in both LXSN and 8E6 following UV in hTERT HFKs (b), but not in primary HFKs? On lines 130–131, the levels of ATM and pATM are said to be lower in 8E6, but this is not consistent with the blot in 2b.
There are a couple of valid points raised by the reviewer in this comment. First, we honestly do not understand why ATM levels decrease after UV in hTERT immortalized cells. It genuinely interests us. We speculate that the answer may lie in the known interactions between telomerase (as well as telomeres more generally) and cellular DNA repair machinery. We now note this observation in the discussion, but unraveling the underlying mechanism would be better addressed in a separate manuscript. Lines 320-324.
The reviewer’s second point (differences between hTERT and primary HFKs) is a reoccurring blunder in our prior submission. Our previous conclusions about HPV 8E6 biology were based on the idea that primary HFKs better reflect β-HPV biology. These cells have rather short lifespans in culture that likely better reflect the equally transient nature of β-HPV infections. Further as mentioned above, telomerase has established interactions with DNA repair protein that can complicate our interpretation of the data. We erred by not sharing this rationale with the reader and have corrected this mistake in the discussion. Lines 321-324.
In figure 2b and c, levels of pBRCA1 are shown. These are not consistent between experimental systems, and only the increase in hTERT cells is described in the text (lines 136–138).
Our statements have been adjusted in accordance with our reply to point 3. Lines 138-141.
On lines 157–185 it is stated that pATR levels did not increase in 8E6 cells. In figure 3b, they do, while in 3c they do not (though there seems to be a difference in the intensity reported and the bands on the blot). The authors could have concluded that the increase was smaller, but not that there was no increase in pATR.
The reviewer is correct and we have incorporated their suggestion. Lines 159-161.
The authors state on lines 175–176 that there is a decrease in total Chk1. In figure 4, the levels of Chk1 are consistently shown to be stable.
Our statements have been adjusted. Lines 176-178
The levels of CDC25A and CDC2 in figure 4E do not appear to show clear patterns of variation, are the authors sure that these findings are reproducible?
Our statements have been adjusted, but this does not diminish the importance of the observation that β-HPV 8E6’s attenuation of ATR/ATM signaling is incomplete. Lines 191-194.
The gating used for cell cycle analysis varies between experimental settings; in particular, the number of cells found to be in the S phase seems clearly related to the size of the S gate. In LXSN cells, delineating cells in the S phase seems very challenging, as the transition from G1 to G2/M is completely smooth. Drawing conclusions based on the amount of S phase cells here seems inappropriate. In addition, the differences between baseline LXSN and 8E6 cells seems large enough that the subtle differences in protein levels described in the study might be influenced by the state of the cells rather than by direct signalling through 8E6. While the direction of causality here could probably be in either direction (cell cycle changes -> ATM/ATR changes, or ATM/ATR changes -> cell cycle changes), the fact that the clear difference in abundance of G2 cells was not reproducible seems to suggest that the experimental system is not stable enough to make any clear inferences about the subtle differences the authors describe.
We have adjusted the gating on our flow cytometry so that it is consistent among the four conditions /cell line permutations. Differences in cell cycle distribution remain, but we have softened our statements in this regard. Lines 195-196.
On lines 222–223, the levels of pXPA and total XPA are said to be lowered by 8E6. For total XPA, the difference is very subtle. For pXPA, there does not seem to be any difference in C or D, and maybe a difference in A. This suggests that the subtle differences seen are not reproducible. There is also a difference between hTERT HFKs and primary HFKs (4–6 hours after UV).
Our statements have been adjusted in accordance with our reply to point 3. Lines 226-227.
On line 246, polη is describes as being lower in 8E6-expressing cells. In figure 6c, the opposite is shown, suggesting that the result is not reproducible.
Our statements have been adjusted in accordance with our reply to point 3. Lines 245-246.
The authors have conducted an in silico analysis, but this is not mentioned in the discussion. Why was the in silico analysis performed? And what, exactly, do the authors conclude based on the in silico analysis?
The in silico analysis helped identify cellular targets likely to be altered by β-HPV 8E6’s attenuation of ATR and ATM expression. We were wrong to have omitted it from the discussion and have corrected this mistake in the resubmission. Lines 279-282.
The analysis of expression dichotomized by mutations in ATM/ATR (suppl fig 3). How do these results “support the role of these kinases as broad regulators of UV responsive gene expression”? A lack of significant associations could be interpreted as evidence that expression of these genes are independent of ATM/ATR activity.
We now make a neutral statement about these data and would be happy to omit them all together. Lines 85-87. ATM/ATR mutations were too rare to produce robust results.
On line 102–104, the authors ask how much the association between ATM/ATR and other genes is due to p53. If they had performed the expression correlation separately in p53-mutated cell lines and p53-wt cell lines, any genes significant in both would presumably not be “regulated” by ATM/ATR through p53. It looks like something like this was the intention of the analysis in table 1, but what has been done here seems to be three separate differential expression analyses. If this is the case, determining whether an association is due to one factor, rather than another is not really that meaningful for finding out what regulates what.
The reviewer is correct and we would love to do this analysis. Unfortunately, and as noted in response to point 12, ATM/ATR mutations were too rare within the data set for the suggested interrogation. We still see value in reporting the comparison provided in table 1, but have added a note that we were unable to do a more rigorous analysis. Our statements have been adjusted in accordance with our reply to point 3. Lines 85-87.
Supplemental figure 4 does not have high enough resolution, and seems to describe pATM in HFKs twice. I can understand that replicate immunoblots are not included in the main text, and even as supplementary information, but given the subtle differences and sometimes inconsistent results presented, including the replicates in the densitometry plots might improve the impression of the reliability of the findings.
A higher resolution version of supplemental figure 4 is now provided.
Round 3
Reviewer 2 Report
The authors have implemented some changes to the manuscript, but overall, it seems to me that the data do not support a consistent role of beta-HPV 8E6 in attenuating ATM and ATR signalling in response to UV damage. There is too much variation in the presented results relative to the subtle changes the authors describe, and in some cases, descriptions of the results are erroneous. While replicate experiments are said to have been performed, the results of these are not presented, precluding assessment of the experimental variability. It also seems like the authors disregard negative results in hTERT cells when they do not fit their hypothesis. While the authors may be correct in doing this, no experimental evidence is presented for their interpretations.
Below are comments (in bold) to previous correspondence (italic).
The manuscript has been improved in some respects, but the inclusion of intensity measures has been done in such a way as to confuse the reader. Immunoblotting, the technique on which this study is based, is a semi-quantitative technique. Some variation in intensity measures is therefore expected. The reported values seem to be quite variable, and only small differences are seen in most cases. In some cases, the results from one blot is reported, when another shows results contradicting the authors’ conclusion. Without more robust measures it seems to me that the data presented is insufficient for the conclusions that are drawn.
We certainly were not trying to confuse or deceive anyone and regret that our efforts gave the reviewer this impression. The quantification of the immunoblots was done in two ways and both were provided in our last submission, but the blurriness of the fourth supplemental figure may have obscured the second normalization. We have also added clarifying statement and caveats to reflect places where uncertainty or inconsistencies exist.
I did not mean to imply that the authors meant to confuse, and certainly not to deceive anyone. The point is simply that there is too much variation in the results for a reader to trust the small differences reported.
Here are my comments for the revised manuscript:
The authors have presented intensity measures for immunoblots. However, when a different baseline is set for each antibody in each cell line, and there is no correction for loading (as far as I can see), the numbers presented are not easily interpretable.
As mentioned above, densitometry data was normalized and presented in two manners. In the main text, we show data normalized to the loading control and then normalized to untreated cells within each cell line. This approach was presented in the primary figures because we wanted to highlight altered responses to UV stimulus. We also provide the data normalized to the loading control and then with both cell lines normalized to untreated vector control (see supplemental figure 4). This is provided as averaged data across each experimental repeat. In general, these analyses are complimentary. When they were not, we now mention and discuss the differences.
While it is good that normalization against the loading control (for the figures in the main text) has been done, this is not mentioned in the text.
In figure 2b, for example, the intensity of ATM and phosphorylated ATM seems to be higher in 8E6-cells prior to UV than in controls (in contradiction to the conclusions of the study), but since both are presented as having an intensity of 1, this is hard to assess.
Please see our response above and also lines 131-132 for changes more specific to this particular example.
Yes, but the point is that you say that ATM levels are lower in 8E6 cells, when they are not; the difference is not “seen more clearly”, the ATM level is not lower at all in hTERT 8E6 cells than in controls, it is higher.
ATM and p-ATM in figure 2: Why does ATM levels decrease in both LXSN and 8E6 following UV in hTERT HFKs (b), but not in primary HFKs? On lines 130–131, the levels of ATM and pATM are said to be lower in 8E6, but this is not consistent with the blot in 2b.
There are a couple of valid points raised by the reviewer in this comment. First, we honestly do not understand why ATM levels decrease after UV in hTERT immortalized cells. It genuinely interests us. We speculate that the answer may lie in the known interactions between telomerase (as well as telomeres more generally) and cellular DNA repair machinery. We now note this observation in the discussion, but unraveling the underlying mechanism would be better addressed in a separate manuscript. Lines 320-324.
The reviewer’s second point (differences between hTERT and primary HFKs) is a reoccurring blunder in our prior submission. Our previous conclusions about HPV 8E6 biology were based on the idea that primary HFKs better reflect β-HPV biology. These cells have rather short lifespans in culture that likely better reflect the equally transient nature of β-HPV infections. Further as mentioned above, telomerase has established interactions with DNA repair protein that can complicate our interpretation of the data. We erred by not sharing this rationale with the reader and have corrected this mistake in the discussion. Lines 321-324.
Maybe the hTERT system is less reliable, but maybe not. Perhaps it is the other system that is closer to being biologically relevant. While your reasoning sounds plausible, no evidence is presented to support one interpretation over another.
In figure 2b and c, levels of pBRCA1 are shown. These are not consistent between experimental systems, and only the increase in hTERT cells is described in the text (lines 136–138).
Our statements have been adjusted in accordance with our reply to point 3. Lines 138-141.
In the last comment, the more reliable results were considered to be from the primary fibroblasts. Now the results from hTERTs are emphasized. This seems inconsistent.
On lines 157–185 it is stated that pATR levels did not increase in 8E6 cells. In figure 3b, they do, while in 3c they do not (though there seems to be a difference in the intensity reported and the bands on the blot). The authors could have concluded that the increase was smaller, but not that there was no increase in pATR.
The reviewer is correct and we have incorporated their suggestion. Lines 159-161.
OK
The authors state on lines 175–176 that there is a decrease in total Chk1. In figure 4, the levels of Chk1 are consistently shown to be stable.
Our statements have been adjusted. Lines 176-178
On lines 178–179, the authors still state that there is a reduced abundance of CHK1.There is no such decrease in figure 4. Or is there something I am missing here?
The levels of CDC25A and CDC2 in figure 4E do not appear to show clear patterns of variation, are the authors sure that these findings are reproducible?
Our statements have been adjusted, but this does not diminish the importance of the observation that β-HPV 8E6’s attenuation of ATR/ATM signaling is incomplete. Lines 191-194.
While I guess it is ok to be surprised by a lack of reduction in CDC25A (or its phosphorylation), this must have been somewhat less surprising than the initially reported increase. This lack of reduction could, as you say, be caused by incomplete inhibition or redundant kinases, but as this is essentially a null result (even a variable one), there is little need to interpret these observations too deeply.
The gating used for cell cycle analysis varies between experimental settings; in particular, the number of cells found to be in the S phase seems clearly related to the size of the S gate. In LXSN cells, delineating cells in the S phase seems very challenging, as the transition from G1 to G2/M is completely smooth. Drawing conclusions based on the amount of S phase cells here seems inappropriate. In addition, the differences between baseline LXSN and 8E6 cells seems large enough that the subtle differences in protein levels described in the study might be influenced by the state of the cells rather than by direct signalling through 8E6. While the direction of causality here could probably be in either direction (cell cycle changes -> ATM/ATR changes, or ATM/ATR changes -> cell cycle changes), the fact that the clear difference in abundance of G2 cells was not reproducible seems to suggest that the experimental system is not stable enough to make any clear inferences about the subtle differences the authors describe.
We have adjusted the gating on our flow cytometry so that it is consistent among the four conditions /cell line permutations. Differences in cell cycle distribution remain, but we have softened our statements in this regard. Lines 195-196.
In the four conditions shown, there is now no difference in S phase abundance. Yet in the discussion, the authors still say there are differences. This is clearly wrong. The authors also say that there were subtle changes to cell cycle distribution (line 197). Which changes?
On lines 222–223, the levels of pXPA and total XPA are said to be lowered by 8E6. For total XPA, the difference is very subtle. For pXPA, there does not seem to be any difference in C or D, and maybe a difference in A. This suggests that the subtle differences seen are not reproducible. There is also a difference between hTERT HFKs and primary HFKs (4–6 hours after UV).
Our statements have been adjusted in accordance with our reply to point 3. Lines 226-227.
There is a reduction in in p-XPA one of the cell lines in two out of six time-points. Apart from that, there is no difference. One should be careful in putting too much into these two observations.
On line 246, polη is describes as being lower in 8E6-expressing cells. In figure 6c, the opposite is shown, suggesting that the result is not reproducible.
Our statements have been adjusted in accordance with our reply to point 3. Lines 245-246.
The reduction is not “more consistent” in primary HFKs, as there is no reduction in hTERT cells.
The authors have conducted an in silico analysis, but this is not mentioned in the discussion. Why was the in silico analysis performed? And what, exactly, do the authors conclude based on the in silico analysis?
The in silico analysis helped identify cellular targets likely to be altered by β-HPV 8E6’s attenuation of ATR and ATM expression. We were wrong to have omitted it from the discussion and have corrected this mistake in the resubmission. Lines 279-282.
OK
The analysis of expression dichotomized by mutations in ATM/ATR (suppl fig 3). How do these results “support the role of these kinases as broad regulators of UV responsive gene expression”? A lack of significant associations could be interpreted as evidence that expression of these genes are independent of ATM/ATR activity.
We now make a neutral statement about these data and would be happy to omit them all together. Lines 85-87. ATM/ATR mutations were too rare to produce robust results.
As you are the authors of this study, inclusion of any results is up to you. If suggestions for additional analyses are not useful, you should say so, rather than including them.
On line 102–104, the authors ask how much the association between ATM/ATR and other genes is due to p53. If they had performed the expression correlation separately in p53-mutated cell lines and p53-wt cell lines, any genes significant in both would presumably not be “regulated” by ATM/ATR through p53. It looks like something like this was the intention of the analysis in table 1, but what has been done here seems to be three separate differential expression analyses. If this is the case, determining whether an association is due to one factor, rather than another is not really that meaningful for finding out what regulates what.
The reviewer is correct and we would love to do this analysis. Unfortunately, and as noted in response to point 12, ATM/ATR mutations were too rare within the data set for the suggested interrogation. We still see value in reporting the comparison provided in table 1, but have added a note that we were unable to do a more rigorous analysis. Our statements have been adjusted in accordance with our reply to point 3. Lines 85-87.
The suggestion here was to split the analysis based on tp53-mutations, not ATM-mutations.
Supplemental figure 4 does not have high enough resolution, and seems to describe pATM in HFKs twice. I can understand that replicate immunoblots are not included in the main text, and even as supplementary information, but given the subtle differences and sometimes inconsistent results presented, including the replicates in the densitometry plots might improve the impression of the reliability of the findings.
A higher resolution version of supplemental figure 4 is now provided.
If you include the replicates in these plots, there is no mention of this in the text. Data from replications should be also visualized in the figure, either using error bars or simply by showing all data points, as this would bolster the reader’s impression of the reproducibility of the results.
Author Response
We thank the reviewer for their time and effort to improve the manuscript. Our detailed responses to their critiques are in the attached word document.
